# The efficacy and safety of prokinetics in critically ill adults receiving gastric feeding tubes: A systematic review and meta-analysis

**Rong Peng**[1,2,3,4], **Hailong Li**[1,2,3], **Lijun Yang**[5], **Linan Zeng**[1,2,3], **Qiusha Yi**[1,2,3], **Peipei Xu**[1,2,3], **Xiangcheng Pan**[1,2,3], **Lingli Zhang**📷[1,2,3]*

1 Department of Pharmacy, West China Second University Hospital, Sichuan University, Chengdu, Sichuan, China, 2 Evidence-Based Pharmacy Center, West China Second University Hospital, Sichuan University, Chengdu, Sichuan, China, 3 Key Laboratory of Birth Defects and Related Diseases of Women and Children (Sichuan University), Ministry of Education, Chengdu, Sichuan, China, 4 Department of Clinical Nutrition, Affiliated Hospital of Chengdu University, Chengdu, Sichuan, China, 5 Department of General Practice Medicine, Affiliated Hospital of Chengdu University, Chengdu, Sichuan, China

* zhanglingli@scu.edu.cn

**Data Availability Statement:** All data generated or analysed during this study are included in this published article.

## Abstract

### Background

Intolerance to gastric feeding tubes is common among critically ill adults and may increase morbidity. Administration of prokinetics in the ICU is common. However, the efficacy and safety of prokinetics are unclear in critically ill adults with gastric feeding tubes. We conducted a systematic review to determine the efficacy and safety of prokinetics for improving gastric feeding tube tolerance in critically ill adults.

### Methods

Randomized controlled trials (RCTs) were identified by systematically searching the Medline, Cochrane and Embase databases. Two independent reviewers extracted the relevant data and assessed the quality of the studies. We calculated pooled relative risks (RRs) for dichotomous outcomes and the mean differences (MDs) for continuous outcomes with the corresponding 95% confidence intervals (CIs). We assessed the risk of bias using the Cochrane risk-of-bias tool and used the Grading of Recommendations Assessment, Development, and Evaluation (GRADE) methodology to rate the quality of the evidence.

### Results

Fifteen RCTs met the inclusion criteria. A total of 10 RCTs involving 846 participants were eligible for the quantitative analysis. Most studies (10 of 13, 76.92%) showed that prokinetics had beneficial effects on feeding intolerance in critically ill adults. In critically ill adults receiving gastric feeding, prokinetic agents may reduce the ICU length of stay (MD -2.03, 95% CI -3.96, -0.10; P = 0.04; low certainty) and the hospital length of stay (MD -3.21, 95% CI -5.35, -1.06; P = 0.003; low certainty). However, prokinetics failed to improve the outcomes of reported adverse events and all-cause mortality.

**Funding:** The study was supported by National Major Science and Technology Projects of China (Award Number: 2017ZX09304029, Recipient: Lingli Zhang), Sichuan Province Science and Technology Major Project (Award Number: 2017JY0067, Recipient: Lingli Zhang), the Major Project of Sichuan health committee (Award Number: 18ZD042, Recipient: Lingli Zhang), the Major Project of Sichuan Province Science and Technology in field of social development (Award Number: 20ZDYF3101, Recipient: Lingli Zhang), Sichuan Science and Technology Program (Award Number: 2020YJ0198, Recipient: Rong Peng), the Project of Education Department of Sichuan Province (Award Number: 18ZB0146, Recipient: Rong Peng), the Project of Chengdu Municipal Health Commission (Award Number: 2020088, Recipient: Rong Peng), the major Project of Affiliated Hospital of Chengdu University (Award Number: 2020YZZ04, Recipient: Rong Peng). The funders had no role in study design, data collection and analysis, decision to publish, or preparation of the manuscript.

**Competing interests:** The authors have declared that no competing interests exist.

## Conclusion

As a class of drugs, prokinetics may improve tolerance to gastric feeding to some extent in critically ill adults. However, the certainty of the evidence suggesting that prokinetics reduce the ICU or hospital length of stay is low. Prokinetics did not significantly decrease the risks of reported adverse events or all-cause mortality among critically ill adults.

## Introduction

Critical illness is usually associated with catabolic stress and increases the incidence of infection and multiple organ dysfunction, resulting in a high mortality rate. A systematic review found a strikingly high prevalence of malnutrition in intensive care unit (ICU) patients (ranging from 38% to 78%) [1]. Owing to the benefits of nutrition support with regard to reducing disease severity and favorably impacting patient outcomes, early nutrition support therapy, primarily by the enteral route, is seen as a proactive therapeutic strategy [2]. In addition, if oral intake is not possible, tube feeding through gastric access has been recommended as the standard approach to initiating enteral nutrition in critically ill adult patients [3].

However, enteral tube feeding intolerance is common in critically ill patients, especially those receiving gastric feeding [3, 4]. Blaser et al. reported that the pooled proportion of feeding intolerance was 38.3% (95% confidence interval (CI) 30.7–46.2%) [4]; besides, a meta-analysis by the European Society for Clinical Nutrition and Metabolism showed that gastric feeding intolerance was more prevalent than postpyloric feeding intolerance (25.7% vs. 3.5%, p = 0.0005) [3]. In addition, feeding intolerance is associated with elevated mortality, and seven-day feeding intolerance is an independent predictor of 60-day mortality [5]. Given the risk associated with gastric feeding intolerance, it should be treated aggressively.

The administration of prokinetics is the method most commonly used to treat gastric feeding intolerance. Among recipients of gastric feeding, 13% had been prescribed prokinetics preemptively before they developed intolerance. Approximately one-third of patients who developed feeding intolerance were treated with a prokinetic agent during their stay in the ICU [6]. Although the use of prokinetics in the ICU is common, guidelines or recommendations are little agreement on how to use prokinetics for gastric feeding intolerance in critically ill patients [2, 3, 7, 8]. One of the reasons for the different recommendations may be that the definition of feeding intolerance has changed over time, especially regarding a high gastric residual volume (GRV). A GRV of 500 mL is the recommended threshold for a diagnosis of enteral feeding intolerance in US and European critical care and nutrition society guidelines [2, 3, 9]. Although the updated European Society for Clinical Nutrition and Metabolism (ESPEN) guidelines [3], published in 2019, provide the latest information on enteral nutrition (EN) and parenteral nutrition (PN) in critically ill adult patients, we find that some aspects of the efficacy and safety of prokinetics in critically ill patients are still unclear [10], and it is necessary to find new evidence to address these uncertainties.

On this topic, a previous meta-analysis by Lewis, K. et al. [11] examined the effects of prokinetics on feeding intolerance or high GRV and clinical outcomes. However, Lewis, K. et al. [11] defined feeding intolerance as GRV $\geq$150 mL, vomiting, or abdominal distention resulting in feeding interruption. This definition may be considered obsolete [12]. Some new evidence has emerged on this topic; therefore, we conducted this systematic review to determine the efficacy and safety of prokinetics for the treatment of gastric feeding intolerance in critically ill adult patients.

## Methods

This systematic review and meta-analysis was conducted according to the Cochrane Handbook for Systematic Reviews of Interventions (version 5.1.0) [13], and the reporting of our study was based on the Preferred Reporting Items for Systematic Reviews and Meta-analyses (PRISMA) statement [14]. The review protocol is available on PROSPERO, registration number CRD42020157446.

Neither patients who received gastric feeding in the ICU nor their families were involved in defining the research question or the outcome measures, but they were involved in the design, providing our team with substantial useful advice regarding design ideas.

### Search strategy

We searched the Medline and Embase databases as well as the Cochrane Central Register of Controlled Trials (CENTRAL) from their inception dates to November 22, 2019. We combined Medical Subject Headings (MeSH) and free text terms to identify relevant articles. An informatics expert developed our search strategies.

We also searched clinicaltrials.gov (https://clinicaltrials.gov/) and the WHO International Clinical Trials Registry Platform (ICTRP) (http://apps.who.int/trialsearch/) for additional information, using the terms "critically ill patients", and limited our search to studies labeled "completed" AND "interventional studies (clinical trials)" in which summary results were available to identify additional eligible studies. There were no language restrictions. Additionally, we used a manual search strategy to retrieve the relevant articles cited by the retrieved publications (the search strategies are reported in **S1 Table**).

### Inclusion criteria

Trials were selected based on the following inclusion criteria: (1) the study was designed as a randomized controlled trial (RCT) comparing prokinetic treatment with a control group; (2) the population included critically ill adult patients aged ≥18 years who were admitted to the ICU and received gastric feeding tubes regardless of whether they had pre-existing feeding intolerance; (3) the intervention group received metoclopramide, erythromycin, or other prokinetic agents, such as herbal medicines or natural medicines intended to enhance gastric motility, regardless of the dose, frequency, duration or combination of prokinetics; (4) the control group received no intervention or a placebo; (5) if the gastric feeding patients with feeding intolerance had a GRV ≥500 mL and/or symptoms of nausea, vomiting, abdominal distention, regurgitation, deterioration in hemodynamics or other symptoms resulting in feeding interruption and failed to respond to interventions, regardless of whether they were in the control group or the prokinetics group, they were switched to postpyloric feeding or had gastric feeding withheld for 4–6 h [2, 3]; and (6) the outcomes included any of the following: all-cause mortality; Acute Physiology and Chronic Health Evaluation II (APACHE II) or Simplified Acute Physiology Score II; sepsis; use of an artificial airway; pneumonia; hospital or ICU length of stay; patient nutritional status (malnutrition); gastrointestinal symptoms; GRV; feeding intolerance; or side effects of the prokinetics, such as cardiovascular disorders, bronchospasm, extrapyramidal symptoms, abdominal cramps, allergic reactions and pancreas disorders. The exclusion criteria were as follows: (1) the studies had no control group; (2) the studies had no prokinetic treatment group; (3) patients were considered to have feeding intolerance if tube feeding was electively not prescribed or was stopped/interrupted for procedural reasons; (4) the studies discontinued or interrupted gastric feeding prematurely when the GRV was less than 500 mL or the patients did not have any signs of intolerance, such as

nausea, vomiting, abdominal pain, abdominal distension, or deterioration in hemodynamics or overall status.

For our purposes, gastric feeding intolerance was defined as a "large" GRV ($\geq$500 mL), the presence of gastrointestinal symptoms (vomiting, diarrhea, gastrointestinal bleeding, the presence of enterocutaneous fistulas), or inadequate delivery of EN (the energy provided by EN was less than 20 kcal/kg BW/day after 72 h of feeding attempts or less than 60% of the EN target on the fifth day) in critically ill adults receiving nutrition via gastric feeding tubes. Preventive usage of prokinetics meant that prokinetics were prescribed preemptively on the day EN was initiated and before patients presented a GRV >150 mL or symptoms of feeding intolerance. Preventive usage of prokinetics for risk meant that prokinetics were used in patients with GRVs between 150 and 500 mL before the development of intolerance. Therapeutic usage of prokinetics meant that the prokinetics were administered to patients who had developed feeding intolerance.

A reported adverse event was defined as any untoward medical occurrence or unfavorable and unintended sign, including an abnormal laboratory finding, symptom, or disease (new or exacerbated), temporally associated with the use of the study medication. The reported adverse events included abnormal laboratory test results (hematology, clinical chemistry, or urinalysis) or other safety assessments (e.g., ECGs, radiological scans, or measurements of vital signs), including those that worsened from baseline and were deemed clinically significant in the medical and scientific judgment of the investigator; exacerbation of a chronic or intermittent preexisting condition, including an increase in the frequency and/or intensity of the condition; new conditions detected or diagnosed after the administration of study medication even if they may have been present prior to the start of the study; and/or signs, symptoms, or clinical sequelae of a suspected interaction, such as diarrhea, nosocomial pneumonia, severe sepsis, brain herniation, cardiac arrest, or changes in the electrocardiographic QTc interval.

### Risk-of-bias assessments

The methodological quality of the included RCTs was assessed independently by 2 researchers (RP, HLL) based on the Cochrane risk-of-bias criteria [13]. The seven items used to evaluate bias in each trial included randomization sequence generation, allocation concealment, blinding of participants and personnel, blinding of outcome assessment, incomplete outcome data, selective reporting, and other bias. We defined other bias as being present in the trials in which the baseline characteristics were not similar between different intervention groups. The included trials were graded as low, unclear, or high risk of bias based on the following criteria: (1) trials were considered high risk of bias if either randomization or allocation concealment was assessed as having a high risk of bias, regardless of the risk of other items; (2) trials were considered low risk of bias when both randomization and allocation concealment were assessed as having a low risk of bias and all other items were assessed as having a low or unclear risk of bias; (3) trials were considered to have unclear risk of bias if they did not meet the criteria for high or low risk of bias.

### Data extraction

Two researchers (RP, HLL) independently extracted the following information from each eligible RCT: (1) general study characteristics: author name, year of publication, numbers of treatment groups and patients, trial registry number, methods for measuring gastric emptying or GRV, and the definition of feeding intolerance; (2) patient characteristics: sex, age, baseline patient information (presence or absence of pre-existing feeding intolerance, APACHE II score and nutritional status, if reported); (3) primary diseases (the medical, surgical, or

neurosurgical conditions of the critically ill patients); (4) interventions: details of the prokinetic treatment group and control group (e.g., dose, frequency, duration and combination of prokinetics for treatment); and (5) outcomes: gastrointestinal symptoms, feeding tolerance, the number of participants with all-cause death, the ICU length of stay, the hospital length of stay, and the number of reported adverse events.

If the trials had more than 2 groups or used factorial designs and could be analyzed using multiple comparisons, we extracted only the information and data of interest reported in the original articles. If a trial had multiple reports, we collated all data into one study. If a trial had reports in both ClinicalTrials.gov and journal publications, we carefully checked data from these two sources for consistency. If outcome data were reported at multiple follow-up points, we used data from the longest follow-up.

## Statistical analysis

The effect of prokinetics on gastrointestinal symptoms and feeding tolerance, main clinical outcomes of all-cause mortality, ICU length of stay, hospital length of stay, and reported adverse events were analyzed. We recorded data on the number of participants with each outcome event by allocated group and recorded the number of participants with compliance and the participant, who was later thought to be eligible or otherwise excluded from treatment or follow-up. Intention-to-treat (ITT) analysis was conducted. ITT analysis is a comparison of the treatment groups that include all patients as originally allocated after randomization regardless of whether treatment was initiated or completed [15]. The CONSORT (Consolidated Standards of Reporting Trials) recommends ITT analysis as standard practice [16].

We performed a meta-analysis to calculate the relative risks (RRs) or absolute risk differences (ARDs) for the dichotomous data and mean differences (MDs) for the continuous data, 95% CIs using the Mantel-Haenszel method and the inverse variance statistical method, respectively. If sufficient data were not available in the published reports or conference abstract, we contacted the authors of the paper. If the raw data were not the mean and standard deviation, the sample mean and standard deviation were estimated from the sample size, median, range and/or interquartile range [17, 18].

We tested for heterogeneity between trials using a standard $Chi^2$ test, and statistical heterogeneity between summary data was evaluated using the $I^2$ statistic. Sensitivity analysis was performed by excluding low-quality studies, trials recruiting participants with particular conditions, or trials with different characteristics from the others. When an inconsistency was detected between the RR and ARD for the same outcome, we explained the results based on the RR because the RR model is more consistent than the ARD, particularly for an intervention aimed at preventing an undesirable event [13, 19].

In our meta-analysis, a random-effects model was used. The defining feature of the random-effects model is that there is a distribution of true effect sizes, and there are two sources of variance, within-study error variance and between-study variance [20]. However, if the number of studies is very small, the statistical power will have poor precision due to the variance between studies. Although the random-effects model is still the appropriate model, the information to apply it correctly is not available. In this case, we will add the separate effects. If heterogeneity was identified ($I^2$ >40% [13]) and sufficient trials were included in the review, we investigated heterogeneity in the specified subgroups based on types of prokinetics (erythromycin, metoclopramide or other prokinetics), the use of a combination of prokinetics (yes or no), and feeding intolerance history (participants with or without pre-existing feeding intolerance before the start of the trial). Analysis was performed to assess whether the difference between the subgroups was statistically significant.

We assessed publication bias by examining funnel plots when the number of trials reporting the primary outcomes was 10 or more. However, if the number of included studies was less than 10 for a given main outcome, the funnel plot may not reliably detect evidence of departure bias. A prototypical situation that should elicit the suspicion of publication bias is when evidence is derived from a small number of studies or small sample sizes and all outcomes favor the intervention [21]. All meta-analyses were performed using RevMan version 5.3 (Cochrane Collaboration). All tests were 2-tailed, and P <0.05 was considered statistically significant.

We used the Grading of Recommendations Assessment, Development, and Evaluation (GRADE) methodology to rate the certainty of evidence as high, moderate, low, or very low. RCTs begin as high-certainty evidence but can be downgraded because of the risk of bias, imprecision, inconsistency, indirectness, or publication bias. If the limitation on the evidence was considered serious, the evidence was downgraded by one level; if the limitation was considered very serious, the evidence was downgraded by two levels [22].

## Results

Our initial search identified a total of 595 citations. After deduplication, 459 publications remained. The titles and abstracts of those records were screened for inclusion, and 48 reports proved potentially eligible. After full-text screening, fifteen trials met the inclusion criteria [23–37]. Five studies did not provide useful data for the quantitative synthesis (meta-analysis) [33–37]. Ultimately, 10 trials were included in the quantitative analysis [23–32]. A total of 846 patients were enrolled in the 10 RCTs, including a variety of critically ill patients with medical, surgical, and neurosurgical conditions. The details of the eligible trials are presented in **Fig 1**. Studies were excluded if they had a different trial design [38–42], a different intervention or a different control [43–61], or a different population [62–64] or had been registered with the Clinical Trials Registry Platform (clinicaltrials.gov or WHO ICTRP) and had been labeled "completed" but the outcomes had not been reported [65–70] (**S2 Table**).

The 15 eligible studies reported the use of 10 prokinetics, including metoclopramide, erythromycin, cisapride, GSK962040, mosapride and herbal medicines or natural medicines intended to enhance gastric motility (Chenxia Sijunzi decoction, ginger, fenugreek seed powder, gastrolit (*Zataria multiflora*), rikkunshito), respectively. Based on the outcomes measured, the studies were subdivided into those investigating effects on gastrointestinal symptoms, feeding tolerance studies, and clinical outcome studies that investigated the hospital length of stay, ICU length of stay, reported adverse events, and all-cause mortality. The details of the eligible studies are presented in **Table 1**.

### Risk of bias

There was one trial with a low risk of bias [24], and two studies had a high risk of bias [25, 28] due to inappropriate randomization and/or allocation concealment. For the remaining 12 studies, we were unable to comprehensively evaluate the risk of bias due to the lack of information [23, 26, 27, 29–37]. (**S1 Fig**).

### Publication bias

We checked the funnel plots of the main outcomes for asymmetry; however, we included fewer than 10 RCTs in each main outcome; therefore, that the funnel plots could not be used to reliably detect departure bias.

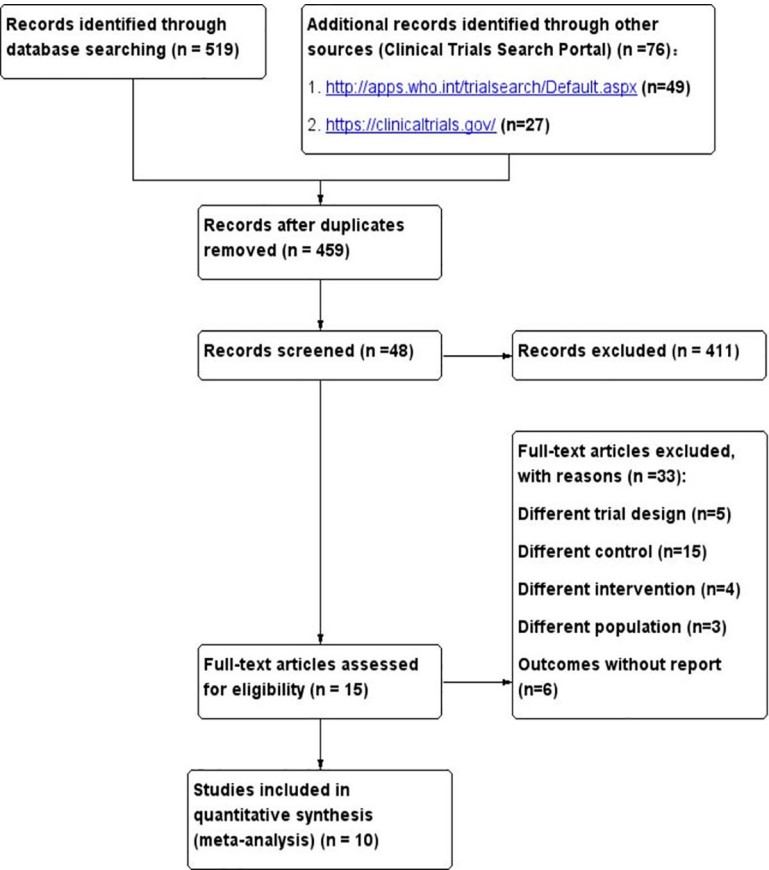

**Fig 1. Literature search and screening process.**

## Main outcomes

**Effect on gastrointestinal symptoms and feeding tolerance.** Thirteen studies evaluated the effect of prokinetics on gastrointestinal symptoms and/or feeding tolerance in adult critically ill patients receiving gastric feeding [23–27, 29–31, 33–37]. The following main results were assessed: gastric emptying, GRV, diarrhea, constipation, feeding complications and feeding intolerance. Gastric emptying was measured by the drug model of acetaminophen absorption or the 13C-octanoic acid breath test with calculation of the gastric emptying time, gastric emptying coefficient or area under the plasma concentration-time curve. The various outcome definitions, especially for gastric tube tolerance, precluded quantitative synthesis of the data.

As a class of drugs, prokinetic agents appear to have positive effects on gastrointestinal function and feeding tolerance. Ten of the thirteen studies reported positive effects on gastric emptying and/or feeding intolerance in critically ill patients who used of prokinetic agents. However, two studies suggested that metoclopramide had no effect on decreasing gastrointestinal complications in adult neurocritical patients or critical traumatic brain injury patients. One study reported that rikkunshito did not improve the achievement of enteral calorie targets in critically ill adults (**Table 2**).

**Effect on hospital or ICU length of stay.** The effect of prokinetics on hospital length of stay was examined by five studies [23, 25–27, 29]. These five studies, which enrolled a total of 250 patients, showed a significant difference in the hospital length of stay between the prokinetic agent-treated group and the control group (MD -3.21, 95% CI -5.35, -1.06; P = 0.003; $I^2$ =

**Table 1. Characteristics of the included trials and participants.**

| Included Trials | Population | Treatment # | Main outcomes | Definition of feeding intolerance † | Prokinetic initiation timing * |
|---|---|---|---|---|---|
| Yavagal et al 2000 (India) [32] | ICU patients required placement of a nasogastric tube for >24 hrs. Mean age: 36.22 years, 61.97% male. Mean APACHE II score: 17.54. | 1) Metoclopramide 10 mg NG q8h; 2) Placebo. | 1) Nosocomial pneumonia; 2) Mortality. | NA | Preventive usage |
| Sustic et al 2005 (Croatia) [36] | Patients treated at a cardiosurgical ICU after CABG surgery, enteral feeding by nasogastric tube. Mean age: 59.5 years, 77.5% male. Mean SAPS II score: 21. | 1) Metoclopramide 10 mg i.v.; 2) Control group. | 1) $t_{+15}$, $t_{+30}$, $t_{+60}$, $t_{+120}$; 2) $AUC_{120}$; 3) $C_{max}$. | NA | Preventive usage |
| Nursal et al 2007 (Turkey) [29] | Traumatic brain injury patients with Glasgow Coma Scale scores of 3–11. Enteral feeding by nasogastric tube. Mean age: 43.42 years, 84.2% male Mean APACHE II score: 12.87. | 1) Metoclopramide 10 mg i.v. q8h×5 days; 2) Control group, saline | 1) Feeding intolerance; 2) Feeding complications; 3) $AUC_{60}$; 4) $C_{max}$; 5) Length of hospital stay; 6) Mortality. | Gastrointestinal symptoms (without GRV) | Preventive usage |
| Nassaji et al 2010 (Islamic Republic of Iran) [28] | Surgical ICU with a nasogastric tube for more than 24 hours. Mean age: 44.88 years, 65.45% male. Mean APACHE II score: not reported. | 1) Metoclopramide 10 mg NG q8h; 2) Control patients did not receive metoclopramide. | 1) Nosocomial pneumonia; 2) Mortality. | NA | Preventive usage |
| Acosta-Escribano et al 2014 (Spain) [23] | Adult neuro-critical patients, Glasgow Coma Scores of 14 to 9 points, with ventilation indications at admission and the need for artificial enteral nutrition. Mean age: 54.53 years, 65.14% male. Mean APACHE II score: 18.53. | 1) Metoclopramide 10 mg; 2) Placebo. | 1) Gastrointestinal complications; 2) Gastric residual; 3) Mechanical ventilation-associated pneumonia; 4) Duration of mechanical ventilation; 5) Length of ICU stay; 6) Length of hospital stay; 7) Mortality. | Large GRV alone (>500 mL in two consecutive episodes) | Preventive usage |
| Rajan et al 2017 (India) [35] | Critically ill cirrhotic patients in a liver ICU with feeding intolerance. | 1) Metoclopramide i.v., 2) Erythromycin i.v., 3) Placebo. | 1) Mortality; 2) GRV. | Gastrointestinal symptoms including large GRV (500 mL) | Therapeutic usage |
| Ritz et al 2005 (Australia) [30] | Mixed medical/surgical intensive care unit patients with mechanic ventilation. Mean age: 47.49 years, 60.9% male. Mean APACHE II score: 19. | **1) Erythromycin 70 mg;** 2) Erythromycin 200 mg; 3) Placebo, saline (0.9%). | 1) Gastric emptying coefficient; 2) Gastric half-emptying time ($t_{1/2}$). | NA | Preventive usage |
| Spapen et al 1995 (Belgium) [31] | Adult medical/surgical intensive care unit patients requiring prolonged mechanical ventilation and enteral feeding. Mean age: 71.10 years, 52.38% male. Mean APACHE II score: not reported. | 1) Cisapride 10 mg q6h; 2) No treatment. | 1) Gastric residual; 2) The mean time at which 50% of the technetium 99m-labeled test meal was eliminated from the stomach ($T_{1/2}$); 3) Mortality. | NA | Preventive usage |
| Heyland et al 1996 (Canada) [33] | Mechanically ventilated patients in trauma and neurosurgery ICUs. Mean age: 53.9 years, 61% male. Mean SAPS score: 9.5. | 1) Cisapride 20 mg, NG; 2) An identical placebo. | 1) $C_{max}$; 2) $AUC_{180}$. | NA | Preventive usage |
| Chapman et al, 2016 (Australia) [24] | Patients undergoing invasive mechanical ventilation in the ICU with nasogastric feeding. Mean age: 44.67 years, 83.33% male. Mean APACHE II score: 18.14. | **1) GSK962040 (50 mg);** 2) GSK962040 (75 mg); 3) Placebo. | 1) Breath test gastric time to half emptying (BTt½); 2) Gastric emptying coefficient; 3) $AUC_{240}$, $AUC_{60}$; 4) $C_{max}$; 5) Adverse events. | Large GRV alone (>200 mL) at least 6 hours after commencing liquid nutrition at ≥ 40 kcal/hr | Preventive usage for risk |

(*Continued*)

**Table 1.** (Continued)

| Included Trials | Population | Treatment # | Main outcomes | Definition of feeding intolerance † | Prokinetic initiation timing * |
|---|---|---|---|---|---|
| Mokhtari et al 2009 (Islamic Republic of Iran) [34] | Adult respiratory distress syndrome (ARDS) ICU patients. | 1) Ginger, 2) Placebo. | 1) Feeding tolerance; 2) Ventilator-associated pneumonia; 3) ICU-free days; 4) Ventilator-free days; 5) Mortality. | Delayed gastric emptying is one of the major reasons for enteral feeding intolerance | Preventive usage |
| Guo JH, et al 2012 (China) [26] | Feeding with enteral nutrition in critically ill patients. Mean age: 59.49 years, 53.33% male. Mean APACHE II score: not reported. | 1) **Traditional Chinese medicine group: Chenxia Sijunzi decoction**; 2) Western medicine group: mosapride dispersible tablets 5 mg and multienzyme tablets NG; 3) Control group: routine symptomatic treatment without any medicines to promote gastrointestinal function. | 1) Time to bowel sound recovery; 2) Gas passage time by anus; 3) Bowel movement time; 4) Days in the hospital. | NA | Preventive usage |
| Kooshki et al 2018 (Iran) [27] | Mechanically ventilated patients, enteral nutrition with nasogastric tube in two intensive care unit centers. Mean age: 56.95 years, 51.67% male. Mean APACHE II score: 23.2. | 1) Fenugreek seed powder 3 g q12h NG; 2) Routine care. | 1) Diarrhea; 2) Constipation; 3) GRV; 4) Respiratory aspiration; 5) Duration of mechanical ventilation; 6) Length of stay in the hospital; 7) Length of stay in the ICU; 8) Mortality. | Gastrointestinal symptoms | Preventive usage |
| Tahershamsi et al 2018 (Iran) [37] | Mechanically ventilated patients hospitalized in ICU. Mean age: 63.06 years, 60.0% male. Mean APACHE II score: No report. | 1) Gastrolit (*Zataria multiflora*) (20 drops) q8h× 4 days; 2) Placebo = water. | 1) GRV. | NA | Preventive usage |
| Doi et al 2019 (Japan) [25] | Critically ill adult patients requiring enteral nutrition by gastric tube for at least 5 days, and all patients were treated with invasive mechanical ventilation. Mean age: 72.84 years, 77.78% male. Mean APACHE II score: 22.82. | 1) **Rikkunshito 5 g** q8h × 5 days; 2) Rikkunshito 2.5 g q8h× 5 days; 3) No rikkunshito (control). | 1) GRV; 2) The percentage of the target enteral calorie intake achieved at the fifth day; 3) The plasma levels of ghrelin; 4) ICU length of stay; 5) Hospital length of stay; 6) Adverse events; 7) Mortality. | Inadequate enteral nutrition/failure to meet the enteral nutrition target at the fifth day (<60%) | Preventive usage |

NG: nasogastric tube feeding; i.v.: intravenous injection; NA: not applicable; $C_{max}$: peak paracetamol plasma levels; AUC: the area under the paracetamol concentration curve; $t_{+15}$, $t_{+30}$, $t_{+60}$, $t_{+120}$: plasma paracetamol concentrations at 15, 30, 60, and 120 minutes after administration of paracetamol and saline or metoclopramide in patients; SAPS, simplified acute physiology score; GRV, gastric residual volume.

# If the trials had more than 2 groups or factorial designs and permitted multiple comparisons, the subgroup in bold font was extracted in this study.

ʃ The study did not provide useful data for meta-analysis.

28%) (**Fig 2**). Three studies evaluated the effect of prokinetics on ICU length of stay in the critical care setting [23, 25, 27]. These three studies, enrolling a total of 186 patients, showed that prokinetic agents appeared to have a positive effect on shortening ICU length of stay (MD -2.03, 95% CI -3.96, -0.10; P = 0.04; $I^2$ = 0%) (**Fig 3**). Additionally, the separate effects of different prokinetics on the ICU length of stay and hospital length of stay are presented in **S3 Table**.

**Table 2. Effects on gastrointestinal symptoms and feeding tolerance.**

| Study | Population (sample size) | Intervention | Outcome | P Value | Conclusions |
|---|---|---|---|---|---|
| Sustic et al 2005 (Croatia) [36] | Cardiosurgical patients after CABG surgery (40) | 1) Metoclopramide 10 mg i.v.; 2) Control group: saline. | $AUC_{120}$; $C_{max.}$ 574±296; 8.51±2.2 429±309; 5.15±2.8 | 0.027; 0.007 | In CABG surgery patients with early enteral feeding, a single dose of intravenous metoclopramide effectively improves gastric emptying. |
| Nursal et al 2007 (Turkey) [29] | Traumatic brain injury patients with Glasgow Coma Scores of 3–11 (19) | 1) Metoclopramide 10 mg i.v. q8h×5 days; 2) Control group: saline. | FI; feeding complications; $AUC_{60}$ at day 5; $C_{max}$ day 5; 4/10 (40%); 5/10 (50%); 589.6±457.8; 15.8±12.9 2/9 (22.2%); 3/9 (33.3%); 560 ±432.9; 12.0±9.9 | NS; NS; NS; NS | The results were unable to reveal any advantage of using metoclopramide in TBI patients. |
| Acosta-Escribano et al 2014 (Spain) [23] | Adult neuro-critical patients, Glasgow Coma Scores of 14 to 9 points (109) | 1) Metoclopramide 10 mg i.v.; 2) Placebo: saline. | Incidence of gastrointestinal complications; Incidence of GRV>500 mL at day 5; 29/58 (50%); 16/58 (28%) 22/51 (45%); 11/51 (22%) | NS; NS | Metoclopramide has no effect on decreasing gastrointestinal complications in adult neuro-critical patients |
| Rajan et al 2017 (India) [35] | Critically ill cirrhotic patients in a liver intensive care unit (72) | 1) Metoclopramide i.v.; 2) Erythromycin i.v.; 3) Placebo. | Resolution of FI; decrease in GRV beyond 24 hrs; the time to restart enteral nutrition (days) 8.7%; no report; 2.61±0.72 24%; no report; 2.20±0.91 no report; no report; 3.47 ±1.29 | 0.026; no report; 0.03 | Early detection and the addition of prokinetics facilitate the resolution of FI in critically ill cirrhotic patients. Erythromycin is safe and superior to metoclopramide for early resolution of gut paralysis in critically ill cirrhotic patients. |
| Ritz et al 2005 (Australia) [30] | Mixed medical/ surgical intensive care unit patients (35) | 1) **Erythromycin 70 mg i.v.;** 2) Erythromycin 200 mg i.v.; 3) Placebo, saline (0.9%). | Gastric emptying coefficient; gastric half-emptying time ($t_{1/2}$) 3.8 (3.3–4.0); 98 (88–112) min 4.0 (3.6–4.2); 86 (75–104) min 2.9 (2.5–3.7); 122 (102–190) min | <0.05; <0.05 | Treatment with 70 and 200 mg of intravenous erythromycin is equally effective in accelerating gastric emptying in critically ill patients. Doses as low as 70 mg (approx. 1 mg/kg) accelerate gastric emptying in critically ill patients, improving the success of enteral feeding. This effect is observed only in patients with delayed gastric emptying. |
| Spapen et al 1995 (Belgium) [31] | Adult medical/surgical intensive care unit patients (21) | 1) Cisapride 10 mg q6h NG; 2) No treatment. | Gastric residual over one week; gastric emptying time $T_{1/2}$; 17.7±8.9 mL; 18±7 min 94.5±33.3 mL; 78±40 min | <0.001; <0.005 | Gastric emptying in critically ill, sedated, and mechanically ventilated patients can be significantly improved by adding cisapride to a routine enteral feeding protocol. |
| Heyland et al 1996 (Canada) [33] | Mixed intensive care unit patients (72) | 1) Cisapride 20 mg; 2) An identical placebo. | Differences (Day 2—Day 1) in $\Delta C_{max}$; $\Delta t_{max}$; $\Delta AUC_{180}$ 49.1±10.7; -40.8±12.0; 5534 ±1349 12.3±7.0; -4.2±10.4; 2832±769 | 0.005; 0.02; 0.09 | Cisapride enhances gastric emptying in critically ill patients |
| Chapman et al, 2016 (Australia) [24] | Mixed intensive care unit patients (33) | 1) **GSK962040 (50 mg) NG;** 2) GSK962040 (75 mg) NG; 3) Placebo. | Baseline vs. post gastric emptying time BTt½; $AUC_{240}$ 0.65 (0.39,0.1.08); 2.50 (1.68,3.72) 1.85 (0.82,4.15); 0.72 (0.39,1.36) 1.21 (0.68,2.15); 1.33(0.85,2.06) | No report; no report | A single enteral dose of camincal (50 mg but not 75 mg), accelerates gastric emptying and increases glucose absorption in feeding-intolerant critically ill patients. |
| Mokhtari et al 2009 (Islamic Republic of Iran) [34] | Critically ill adult respiratory distress syndrome (ARDS) patients (32) | 1) Ginger NG, 2) Placebo. | Feeding tolerated in the first 48 hrs; feeding tolerated during the entire study period 51%; 92% 57%; 93% | <0.005; 0.42 | Supplementing the diet with ginger extract in ARDS patients reduces the delayed gastric emptying risk. |

*(Continued)*

**Table 2.** (Continued)

| Study | Population (sample size) | Intervention | Outcome | P Value | Conclusions |
|---|---|---|---|---|---|
| Guo JH, et al 2012 (China) [26] | Mixed intensive care unit patients (80) | 1) **Traditional Chinese medicine group: Chenxia Sijunzi decoction;** 2) Western medicine group: mosapride dispersible tablets 5 mg and multienzyme tablets NG; 3) Control group: routine symptomatic treatment without any medicines to promote gastrointestinal function. | The time to bowel sound recovery; the time to passage of gas by anus recovery; the time to bowel movement recovery 41.02±7.52[a]; 49.90±6.95[a]; 58.22±6.71[a] 44.02±6.23[a]; 51.32±5.12[a]; 60.91±3.72[a] 54.62±5.51; 64.68±9.47; 78.20±7.11 | [a]P<0.01 | Chenxia Sijunzi decoction can promote severe patients' gastrointestinal function recovery. No significant differences in each testing index were found between the traditional Chinese medicine and Western medicine groups. |
| Kooshki et al 2018 (Iran) [27] | Mixed intensive care unit patients (60) | 1) Fenugreek seed powder 3 g q12h NG; 2) Routine care. | GRV at the 5th day; diarrhea; constipation; respiratory aspiration at 5th/6th days 28.06±9.23; 1/30 (3.3%); 3/30 (10%); 1/30 (3.3%) 38.94±9.54; 6/30 (20%); 21/30 (70%); 10/30 (33.3%) | 0.001; 0.04; 0.001; 0.005 | Beneficial effects of fenugreek seeds on food intolerance were observed in critically ill patients. |
| Tahershamsi et al 2018 (Iran) [37] | Mixed intensive care unit patients (50) | 1) Gastrolit (Zataria multiflora) (20 drops) q8h× 4 days; 2) Placebo = water. | GRV on the second, third, and fourth days The data could not be extracted | All P<0.0001 | Gastrolit can decrease the GRV in mechanically ventilated patients |
| Doi et al 2019 (Japan) [25] | Mixed intensive care unit patients | 1) **Rikkunshito 5 g** q8h ×5 days;2) Rikkunshito 2.5 g q8h×5 days;3) No rikkunshito (control). | GRV; the percentage of the target energy at the 5th day; the target energy was achieved at the 5th day No report; 62%; 63% No report; 40%; 38% No report; 59%; 56% | NS; NS; NS | Standard- or high-dose rikkunshito did not improve achievement of the enteral calorie target in critically ill adults. |

FI, feeding intolerance; NS, not significant

[a]P<0.01 compared with the control group.

**Effect on reported adverse events.** Seven studies reported events that met the definition of adverse events in 757 critically ill patients [23–25, 27–29, 32]. The meta-analysis showed no significant difference in the risk of reported adverse events between the prokinetic agent group and the control group (RR 1.13, 95% CI 0.92, 1.38; P = 0.25; $I^2$ = 0%) (**S2 Fig**).

**Effect on all-cause mortality.** The effect of prokinetic agents on all-cause mortality was examined by six studies in 691 critically ill patients [23, 25, 28, 29, 31, 32]. There was no significant difference in all-cause mortality between the prokinetic agent group and the control group (RR 0.96, 95% CI 0.81, 1.14; P = 0.64; $I^2$ = 0%) (**S3 Fig**).

## Subgroup analysis

Although no significant heterogeneity was found, we performed subgroup analysis to determine whether important subgroup differences existed. In the subgroup analysis stratified by

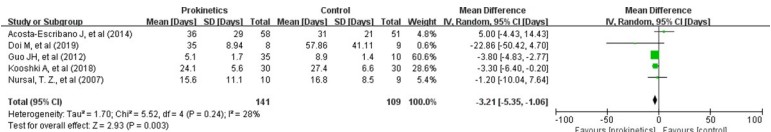

**Fig 2. Forest plot for hospital length of stay outcomes.** IV: inverse variance; CI: confidence interval.

**Fig 3. Forest plot for ICU length of stay outcomes.** IV: inverse variance; CI: confidence interval.

type of prokinetic agents, no significant subgroup differences were detected in the clinical outcomes of hospital length of stay, ICU length of stay, reported adverse events and all-cause mortality (**S4**–**S7 Figs**). Furthermore, no study compared the use of a combination of prokinetics to placebo or no treatment. Only one study about the preventive usage of prokinetics for risk patients investigated reported adverse events [24]. The other studies investigated the preventive usage of prokinetics in all patients. The subgroup analysis of the preventive usage of prokinetics in all patients did not show important changes in the pooled effects of the reported adverse events.

## Sensitivity analysis

The sensitivity analysis, which was performed by excluding the trials with a high risk of bias [25, 28], did not show important changes in the pooled effects of hospital length of stay, ICU length of stay, reported adverse events, and all-cause mortality. (**S8**–**S11 Figs**).

## Certainty of evidence

The certainty of evidence was moderate for the clinical outcome of all-cause mortality. However, the certainty of evidence was low for the clinical outcomes of ICU length of stay, hospital length of stay and reported adverse events. The details of the risk of bias and quality assessment are outlined in **Table 3**.

## Discussion

In this systematic review, we conducted a comprehensive literature search and used objective study inclusion criteria. Fifteen studies were included in the final analysis. Because of the small sample sizes and a relatively small number of eligible studies, the pooled effects are lacking in accuracy in the quantitative analysis. Most studies (10 of 13, 76.92%) showed that prokinetic agents had beneficial effects on feeding tolerance in critically ill adults. The studies that did not show beneficial effects (3 of 13, 23.08%) investigated special populations of neuro-critical patients and critical traumatic brain injury patients taking metoclopramide and rikkunshito. Furthermore, the use of prokinetic agents in critically ill patients receiving gastric feeding may reduce the ICU or hospital length of stay, but the certainty of evidence was low due to the risk of bias and imprecision. Prokinetics did not significantly reduce the risks of reported adverse events or all-cause mortality.

In this study, we examined the effect of prokinetic agents on gastrointestinal symptoms, feeding tolerance and clinical outcomes. Compared to the control group, the group receiving prokinetics did not have a low risk of mortality; these results were the same as the results of the meta-analysis by Lewis, K. et al. [11], but our methods were different. Lewis, K. et al. [11] defined feeding intolerance as either GRV ≥150 mL, vomiting, or abdominal distention resulting in feeding interruption. This definition may be considered obsolete [12]. We defined gastric feeding intolerance as either a GRV ≥500 mL or concomitant symptoms of nausea, vomiting, abdominal distention, regurgitation or other symptoms resulting in feeding interruption in critically ill adult patients with gastric feeding tubes. We excluded studies that

**Table 3. GRADE evidence profile of the efficacy and safety of prokinetics in critically ill adult patients receiving gastric feeding tubes.**

| No. of studies | Study design | Certainty assessment | | | | | No. of patients | | Effect | | Certainty | Importance |
|---|---|---|---|---|---|---|---|---|---|---|---|---|
| | | Risk of bias | Inconsistency | Indirectness | Imprecision | Other considerations | Prokinetics | Placebo | Relative (95% CI) | Absolute (95% CI) | | |
| **Effect on ICU length of stay** | | | | | | | | | | | | |
| 3 | randomized trials | serious[1] | not serious[2] | not serious[3] | serious[4] | none[5] | 96 | 90 | - | MD **2.03 days lower** (3.96 **days lower** to 0.1 **days lower**) | ⊕⊕◯◯ LOW | IMPORTANT |
| **Effect on hospital length of stay** | | | | | | | | | | | | |
| 5 | randomized trials | serious[6] | not serious[7] | not serious[3] | serious[8] | none[5] | 141 | 109 | - | MD **3.21 days lower** (5.35 **days lower** to 1.06 **days lower**) | ⊕⊕◯◯ LOW | IMPORTANT |
| **Effect on reported adverse events** | | | | | | | | | | | | |
| 7 | randomized trials | serious[9] | not serious[10] | not serious[3] | serious[11] | none | 105/320 (32.8%) | 120/437 (27.5%) | **RR 1.13** (0.92 to 1.38) | **40 more RAE per 1,000 patients** (from 20 fewer RAE to 100 more RAE) | ⊕⊕◯◯ LOW | IMPORTANT |
| **Effect on all-cause mortality** | | | | | | | | | | | | |
| 6 | randomized trials | serious[12] | not serious[13] | not serious[3] | not serious[14] | none | 114/286 (39.9%) | 174/405 (43.0%) | **RR 0.96** (0.81 to 1.14) | **30 fewer deaths per 1,000 patients** (from 100 fewer deaths to 40 more deaths) | ⊕⊕⊕◯ MODERATE | CRITICAL |

CI: confidence interval; RR: risk ratio; MD: mean difference; RAE: reported adverse events.

1. We downgraded the quality of evidence for risk of bias by one level. Two of three included studies had a high or unclear risk of bias.

2. We did not downgrade the quality of evidence for inconsistency, $I^2 = 0\%$ and $Chi^2 = 1.45$, $P = 0.48$.

3. Although the studies included any critically ill patient, we did not downgrade for indirectness.

4. We downgraded the quality of evidence for imprecision by one level because the total population size is less than 400. The 95% confidence interval contained a small benefit that did not meet the clinical decision threshold (min. one day).

5. We did not downgrade for publication bias, although we could not assess this category reliably due to the small number of eligible studies. Not all included studies showed benefits of the studied intervention.

6. We downgraded the quality of evidence for risk of bias by one level. Most studies had an unclear risk of bias. In addition, one study lacked allocation concealment and blinding.

7. We did not downgrade the quality of evidence for inconsistency, $I^2 = 28\%$ and $Chi^2 = 5.52$, $P = 0.24$.

8. We downgraded the quality of evidence by one level for imprecision because the population size is less than 400.

9. We downgraded the quality of evidence for risk of bias by one level. Most studies had an unclear risk of bias. In addition, two studies lacked allocation concealment and/or blinding.

10. We did not downgrade for inconsistency, $I^2 = 0\%$ and $Chi^2 = 1.64$, $P = 0.95$.

11. We downgraded the quality of evidence for imprecision by one level because the 95% confidence interval around the pooled effect included both no effect and appreciable harm (a relative risk increase greater than 25%).

12. We downgraded the quality of evidence for risk of bias by one level. Most studies had an unclear risk of bias. In addition, two studies lacked allocation concealment, and two studies lacked blinding.

13. We did not downgrade for inconsistency, $I^2 = 0\%$ and $Chi^2 = 4.52$, $P = 0.48$.

14. We did not downgrade for imprecision because the 95% confidence interval around the pooled effect did not include both no effect and an appreciable benefit (a relative risk reduction greater than 25%) or appreciable harm (a relative risk increase greater than 25%).

discontinued or interrupted gastric feeding prematurely following the disappearance of gastric feeding intolerance. Our meta-analysis included new studies that used this latest definition [23–27, 30, 31], and we identified 5 studies that investigated the administration of prokinetics, including herbal medicines/natural medicines, to critically ill adult patients with gastric feeding tubes [25–27, 34, 37].

Additionally, we found that prokinetic agents might reduce the ICU or hospital length of stay for critically ill patients receiving gastric feeding. However, the number of studies and the sample size were very small, and the certainty of the evidence was low. Furthermore, no significant difference was found between the prokinetic agent groups and placebo/no treatment groups with regard to the risks of reported adverse events and all-cause mortality. Therefore, we cannot draw a convincing conclusion that the use of prokinetics can improve clinical outcomes in critically ill adults. We recommend that more research should be conducted in this field.

This study has several limitations. First, 21 published original studies or trials registered in the International Clinical Trials Registry Platform (WHO) or clinicaltrials.gov were identified. However, 6 trials, although completed, did not have available results, which might have led to the omission of trials meeting the inclusion criteria, resulting in publication bias. Second, some included trials did not record the baseline status of feeding intolerance for all participants. The subgroup results might have been different if all individuals were evaluated. Third, we were unable to comprehensively evaluate the risk of bias in 12 studies due to a lack of information. Fourth, for each outcome, the total sample size was relatively small, which likely resulted in inadequate power to detect a difference in treatment effect. We recommend that more original studies on this topic be conducted.

## Conclusion

As a class of drugs, prokinetic agents may improve gastric feeding tolerance in critically ill adults. However, the certainty of the evidence suggesting that prokinetic agents are effective at reducing the ICU or hospital length of stay is low. There was also no significant reduction in the risk of reported adverse events or all-cause mortality. Additional RCTs are needed to determine the effect of prokinetics on clinical outcomes in critically ill patients in the future.

## Supporting information

**S1 Table. Search strategy.**
(DOCX)

**S2 Table. Excluded studies.**
(DOCX)

**S3 Table. Separate effects of different prokinetic agents on hospital or ICU length of stay.**
(DOCX)

**S4 Table. PRISMA checklist.**
(DOCX)

**S1 Fig. Risk of bias.**
(DOCX)

**S2 Fig. Reported adverse event outcomes.**
(DOCX)

**S3 Fig. All-cause mortality outcomes.**
(DOCX)

**S4 Fig. Subgroup analysis by the type of prokinetic agents for hospital length of stay outcomes.**
(DOCX)

**S5 Fig. Subgroup analysis by the type of prokinetic agents for ICU length of stay outcomes.**
(DOCX)

**S6 Fig. Subgroup analysis by the type of prokinetic agents for reported adverse event outcomes.**
(DOCX)

**S7 Fig. Subgroup analysis by the type of prokinetic agents for all-cause mortality outcomes.**
(DOCX)

**S8 Fig. Sensitivity analysis of hospital length of stay outcomes.**
(DOCX)

**S9 Fig. Sensitivity analysis of ICU length of stay outcomes.**
(DOCX)

**S10 Fig. Sensitivity analysis of reported adverse event outcomes.**
(DOCX)

**S11 Fig. Sensitivity analysis of all-cause mortality outcomes.**
(DOCX)

## Acknowledgments

We are grateful to all the staff in this study for their teamwork and persistent efforts, and we are also thankful to the Group of People with Highest Risk of Drug Exposure of the International Network for the Rational Use of Drugs, China, and the Evidence-Based Pharmacy Committee of the Chinese Pharmaceutical Association for providing methodological advice.

## Author Contributions

**Data curation:** Rong Peng, Hailong Li, Lijun Yang, Lingli Zhang.

**Formal analysis:** Rong Peng, Hailong Li, Lingli Zhang.

**Funding acquisition:** Lingli Zhang.

**Investigation:** Hailong Li.

**Methodology:** Rong Peng, Hailong Li, Lingli Zhang.

**Project administration:** Lingli Zhang.

**Resources:** Rong Peng.

**Supervision:** Linan Zeng, Lingli Zhang.

**Visualization:** Hailong Li, Linan Zeng, Lingli Zhang.

**Writing – original draft:** Rong Peng, Hailong Li, Lijun Yang, Linan Zeng, Qiusha Yi, Peipei Xu, Xiangcheng Pan, Lingli Zhang.

**Writing – review & editing:** Rong Peng, Hailong Li, Lijun Yang, Linan Zeng, Qiusha Yi, Peipei Xu, Xiangcheng Pan, Lingli Zhang.

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
