## [Decision Letter · Decision Letter 0]

19 Mar 2020

PONE-D-20-01167

The efficacy and safety of prokinetics in adult critically ill patients receiving gastric enteral nutrition: a systematic review and meta-analysis

PLOS ONE

Dear Dr. Zhang,

Thank you for submitting your manuscript to PLOS ONE. After careful consideration, we feel that it has merit but does not fully meet PLOS ONE’s publication criteria as it currently stands. Therefore, we invite you to submit a revised version of the manuscript that addresses the points raised during the review process.

Both reviewers asserted that the manuscript was not technically sound, and the conclusions were not supported by the data, and both reviewers found that the text required further editing to be clearly understandable. Please see the reviewer comments for details. This manuscript would require quite a bit of revision and clarification to be suitable for further peer review and potential publication.

Specific highlights of the reviewer comments include:

Although the updated definition of feeding intolerance provides a rationale for a new examination of the question, it appears from the reviewer comments that studies without a clear definition were included (eg, Tahershamsi 2018). Please clarify why these were selected for inclusion.It also appears from the reviewer comments that several relevant studies may have been missed. This issue could be clarified by specifying the references and exclusion criteria for the 27 studies excluded at full text. This could be done in a separate online table if necessary.The reviewer who noted the inclusion of hospital length of stay in the Results but not in the Methods is correct. Please clarify this.Pay attention to reviewer comments highlighting the problem with combining different interventions in the meta-analysis. For example, in the herbal group of studies, why does it make sense to combine rikkunshito and fenugreek, and why does it make sense to produce a summary estimate across all the different types of interventions? Please clarify or revise.The reviewer comments about GRADE evaluations should be responded to. For example, imprecision is present in several analyses based on rule-of-thumb criteria for sample size (eg, at least 400 participants or 300 events in an analysis), as well as potentially based on width of the confidence interval including both benefit and lack of effect. The criteria for decisions about GRADE elements should be specified in the Methods section, and the rationale for decisions should be made clear through footnotes to Table 2.  Regarding publication bias and GRADE, it is appropriate that funnel plots were not created however the rationale for strongly suspecting publication bias in all estimates should be clarified.

In addition, I have the following comments based upon my reading of the manuscript submission:

Under funding disclosure, it is unclear whether the review was funded, and if so by whom. This needs to be specified.The Plos ONE guidelines for systematic reviews state that: ‘Authors must also state in their “Methods” section whether a protocol exists for their systematic review, and if so, provide a copy of the protocol as supporting information and provide the registry number in the abstract.‘ Please include this information.The review Methods should include a description of the procedures for screening of titles/abstracts and full texts.The I-squared is not an evaluation of the statistical heterogeneity between summary data. It is an estimate of the percentage of variability in a meta-analysis that is attributable to study heterogeneity.The choice of a random vs fixed effects model should not be based on heterogeneity according to the Cochrane Handbook and other sources. For more information on choice of models see page 105 of  https://www.meta-analysis.com/downloads/Intro_Models.pdf.

I am sorry that we cannot be more positive on this occasion, but hope that you appreciate the reasons for this decision.

Yours sincerely,

Lisa Susan Wieland

Academic Editor

PLOS ONE

"The study was supported by the Science and Technology Department of Sichuan Province Major

397 Project (Grant no. 2017JY0067), the National Science and Technology Major Project (Grant no.

398 2017ZX09304029) and the Major Project of Sichuan health committee (18ZD042), and the Program

399 for Yangtze River Scholars and Innovative Research Team in University (No. IRT0935).

400 Role of the Funder: The Science and Technology Department of Sichuan Province Major Project, the

401 National Science and Technology Major Project, and the Program for Yangtze River Scholars and

402 Innovative Research Team had no role in the design and conduct of the study; collection,

403 management, analysis, and interpretation of the data; preparation, review, or approval of the

404 manuscript; and decision to submit the manuscript for publication.".

i) We note that you have provided funding information that is not currently declared in your Funding Statement. However, funding information should not appear in the Acknowledgments section or other areas of your manuscript. We will only publish funding information present in the Funding Statement section of the online submission form.

ii) Please remove any funding-related text from the manuscript and let us know how you would like to update your Funding Statement. Currently, your Funding Statement reads as follows:

"The funders had no role in study design, data collection and analysis, decision to publish, or preparation of the manuscript.".

Reviewers' comments:

Reviewer's Responses to Questions

We would appreciate receiving your revised manuscript by May 03 2020 11:59PM. To enhance the reproducibility of your results, we recommend that if applicable you deposit your laboratory protocols in protocols.io, where a protocol can be assigned its own identifier (DOI) such that it can be cited independently in the future. For instructions see: http://journals.plos.org/plosone/s/submission-guidelines#loc-laboratory-protocols

We look forward to receiving your revised manuscript.

Kind regards,

Lisa Susan Wieland

Academic Editor

PLOS ONE

Journal Requirements:

Additional Editor Comments (if provided):

Reviewers' comments:

Reviewer's Responses to Questions

**Comments to the Author**

1. Is the manuscript technically sound, and do the data support the conclusions?

Reviewer #1: No

Reviewer #2: No

2. Has the statistical analysis been performed appropriately and rigorously? 

Reviewer #1: Yes

Reviewer #2: I Don't Know

3. Have the authors made all data underlying the findings in their manuscript fully available?

Reviewer #1: Yes

Reviewer #2: Yes

4. Is the manuscript presented in an intelligible fashion and written in standard English?

Reviewer #1: No

Reviewer #2: No

5. Review Comments to the Author

Reviewer #1: Paragraph 115-118 needs revision. Fragment?

Line 142: end of sentence “ changes of electrocardiogram QTc interval, and so on”. Perhaps removing “and so on” will sound better or can refer to outcomes listed in the paragraph of “inclusion criteria”.

Lines 252-254: for consistency, report results fully with point estimates and 95% CIs.

Lines 256-259: what do the authors mean by “the total situation”. “And there was also no significant difference in the risk of the reported adverse events between prokinetics treatment and the control group on the total situation (RR, 1.18, 95%CI, 0.79 to 1.75, P=0.42; I2=69%).

Lines 264: can the authors explain why they deviated from their data analysis plan of conducting publication bias assessment when 10 or more studies were available (lines 181-183) but lowered the quality of evidence because publication bias was strongly suspected?

Line 271-272: “Subgroup analysis by type of prokinetics.” Fragment.

Line 275-277: When Metoclopramide did not show significant difference in the hospital length of stay compared with the control group (MD, 1.70, 95%CI, -4.75 to 8.15, P=0.61; I2=0%) (S1 Fig). Check grammar please.

Lines 283-284: repetition of 254-256.

Lines 285-286: review grammar.

Line 286-287: pooled effects for what outcome(s)?

Line 288: I think the authors mean to have the first part as a subheading “Subgroup analysis by preventive giving prokinetics or therapeutic usage of prokinetics.”

Would recommend: preventive use instead of preventive giving.

Also would recommend: “pooled effects for primary outcomes” instead of “pooled effects” at the end of sentences.

Lines 298-301: the authors deviate from statistical plan for publication bias again by downgrading for string suspicion of publication bias. Revision needed. Also in GRADE evidence profile confidence intervals for all-cause mortality, ICU and hospital length of stay and adverse events all include significant benefits and harms and yet no downgrading for imprecision was performed. Can the authors elaborate on this ? ¬¬¬For length of stay what was the unit of measurement? Days or hours? This would be important for the outcome of hospital length of stay with natural medicines. If the difference was 9 days fewer and between 18.5 and 0.8 days lower, one might consider downgrading for imprecision specially that the number of patients in that analysis is 122 only. I would strongly recommend the authors consult someone with more experience using the GRADE method.

Lines 315-318: And there was a systematic review about the definition, prevalence, and outcome of feeding intolerance in intensive care, and the result showed that there were more than 60% (40/63) of studies defining the large GRVs with the threshold less than or equal to 250 ml and about 41.7% (30/72) of studies with the sole GRVs levels to definite of feeding intolerance. Not clear and needs to be rewritten.

Lines 325-327: So, we conducted this updated systematic review and meta-analysis, we defined the gastric enteral nutrition feeding intolerance as either GRV ≥500 ml or GRV <500 ml concomitant with symptoms of nausea, vomiting, abdominal distention, regurgitation or other symptoms resulting in feeding interruption. This is not true. Only 3 of the included studies explicitly reported the cutoff for the GRV to be ≥ 500 ml (24, 31, 34) and 1 study reported > 200 ml at any time or > 500 ml/24 hours). I find this misleading.

Line 342: Lewis, K. et al [12] defined feeding intolerance as either GRV ≥150 ml, vomiting, or abdominal distention resulting in feeding interruption. This definition may be considered obsolete [13]. This is very difficult to justify given the point above when the exact amount of GRV was not reported in more than half of the studies included.

Lines 342-347 are exact repetition of lines 324-329.

Line 351: “trails” change to “trials”

Lines 352-358: I disagree with the conclusion. The heterogeneity was 95%, the confidence intervals we extremely wide (especially if this was in days) and the total number of patients is 122. This makes it extremely difficult to make any sound conclusions, contrary to the way reported by the authors.

Lines 380: I am concerned that the while the title of the ,manuscript is prokinetics, the bulk of discussion section and the conclusion is about herbal medicines as prokinetic agents. As mentioned earlier, the authors claim their cutoff for GRV was 500 ml but they included many studies without a clear cutoff and by doing that they missed many studies in Lewis et al. that included generic medicines. In addition, the authors did not include hospital length of stay as an outcome in their methods section (lines 92-101) and yet they perform this analysis and include the forest plot for it in the main manuscript.

The methods section very well written, however the other sections need significant revision of language and grammar.

Reviewer #2: The authors made a new metaanalysis on the efficacy and safety of prokinetic agents administered during enteral feeding in intensive care patients. Previous metaanalysis were already performed on thIs topic. Originality of this work may come from a revised definition for digestive feeding intolerance, which indeed corresponds to more recent experts recommendations (Gastric residual volume (GRV) above 500 mls). It has to be noted however, that even the treshold of 500 mls GRV for definition of gastric enteral feeding intolerance is challenged, and some authors now recommend against measuring any GRV during enteral nutrition in ICU patients.

I have two major concerns with that study:

- Subgroup analysis: what is the rationale of pooling studies in which different substances are administered: indeed, natural/herbal substances are not identical in these pooled studies.

- To my opinion, the conclusion (administration of herbal medicines/natural medicines may reduce hospital length of stay) is not supported by the (meta)analysis. The purpose of a metaanalysis is to pool pertinent studies in order to answer a well predefined question. The conclusion that is drawn by the authors is not really related to the initial question.

6. PLOS authors have the option to publish the peer review history of their article (what does this mean?). If published, this will include your full peer review and any attached files.

Reviewer #1: No

Reviewer #2: Yes: Alain-Michel Dive, CHU UCL Namur, Belgium

---

## [Author Response · Author response to Decision Letter 0]

28 Apr 2020

Dear Editors and Reviewers,

Thank you for your letter and the comments concerning our manuscript (Manuscript Number: PONE-D-20-01167). Those comments are all valuable and very helpful for revising and improving our paper, as well as the important guiding to our researches. We have studied comments carefully and have made correction which we hope meet with approval. Revised portion are marked with the changes highlighted in yellow in the paper. The main corrections in the paper and the responds to the reviewer’s comments are as flowing:

Editors' comments:

We appreciate your generous help and your patience. We take all the comments into consideration, and revise our manuscript accordingly. We hope the revised manuscript can meet with your approval. The answers to your comments are as flowing:

1. Although the updated definition of feeding intolerance provides a rationale for a new examination of the question, it appears from the reviewer comments that studies without a clear definition were included (eg, Tahershamsi 2018). Please clarify why these were selected for inclusion.

Answer: In our study, except for feeding tolerance, the main outcomes included gastrointestinal symptoms and clinical outcomes (hospital length of stay, ICU length of stay, reported adverse events, all-cause mortality). The included studies were not based on whether the study with a clear definition of feeding intolerance. But we excluded these studies that discontinued or interrupted the gastric feeding prematurely when the GRV was less than 500 mL or the patients did not have any signs of intolerance. 

2. It also appears from the reviewer comments that several relevant studies may have been missed. This issue could be clarified by specifying the references and exclusion criteria for the 27 studies excluded at full text. This could be done in a separate online table if necessary.

Answer: all the literatures had been checked again. The inclusion and exclusion studies all met the predefined criteria. And according to your advice, we added a table to explain the exclusion reason for the 27 studies. (Supp. Table 2).

3. The reviewer who noted the inclusion of hospital length of stay in the Results but not in the Methods is correct. Please clarify this.

Answer: we are very appreciated of your kindly advise. Because of our carelessness, “hospital length of stay” did not present in the Inclusion Criteria of Methods section. But the outcome of “hospital length of stay” have already been included in our plan. In the Methods section of the Data Extraction and Statistical Analysis, the outcome of “hospital length of stay” had been included. So, we amended the Methods section in manuscript. we added the outcome of “hospital length of stay” in the Inclusion Criteria of Methods section. 

4. Pay attention to reviewer comments highlighting the problem with combining different interventions in the meta-analysis. For example, in the herbal group of studies, why does it make sense to combine rikkunshito and fenugreek, and why does it make sense to produce a summary estimate across all the different types of interventions? Please clarify or revise.

Answer: According to your advice, we have amended the relevant part in manuscript. In the revised manuscript, we removed the results of the combining herbal group in the meta-analysis. However, as a class of drugs, the results of the combining different types of prokinetics in the meta-analysis were reserved. Although no significant heterogeneity was found, we performed subgroup analyses to determine if there are important subgroup differences.

5. The reviewer comments about GRADE evaluations should be responded to. For example, imprecision is present in several analyses based on rule-of-thumb criteria for sample size (eg, at least 400 participants or 300 events in an analysis), as well as potentially based on width of the confidence interval including both benefit and lack of effect. The criteria for decisions about GRADE elements should be specified in the Methods section, and the rationale for decisions should be made clear through footnotes to Table 2. 

Answer: we felt sorry for these mistakes. We seriously conducted the GRADE evaluation again. In the revised manuscript, the details of the risk of bias and quality assessment were outlined in Table 3, the rationale for decisions had been made clear through footnotes to Table 3. 

6. Regarding publication bias and GRADE, it is appropriate that funnel plots were not created however the rationale for strongly suspecting publication bias in all estimates should be clarified.

Answer: In view of the fact that publication bias arises when investigators fail to report studies they have undertaken (typically those that show no effect). A prototypical situation that should elicit suspicion of publication bias occurs when published evidence is limited to a small number of trials, all of which are showing benefits of the studied intervention. 

In the revised manuscript, the meta-analysis showed that the use of prokinetic agents in critically ill patients receiving gastric feeding may reduce ICU or hospital length of stay, there was no significant difference in the incidence of reported adverse events and all-cause mortality between prokinetic agent group and control group. Although we could not assess this category reliably due to small number of eligible studies, we did not downgrade the quality of evidence for publication bias. Because not all of included studies were showing benefits of the studied intervention in clinical outcome of ICU length of stay and hospital length of stay, respectively.

7. Under funding disclosure, it is unclear whether the review was funded, and if so by whom. This needs to be specified.

Answer: according to your advice, we have amended the relevant part in manuscript. 

8. The Plos ONE guidelines for systematic reviews state that: ‘Authors must also state in their “Methods” section whether a protocol exists for their systematic review, and if so, provide a copy of the protocol as supporting information and provide the registry number in the “abstract”. Please include this information.

Answer: we have registered our protocol in https://www.crd.york.ac.uk/prospero/,

but the review is still ongoing (ID 157446).

9. The review Methods should include a description of the procedures for screening of titles/abstracts and full texts.

Answer: according to your advice, we have amended the relevant part in manuscript. The details of the eligible trials are presented in Figure 1. Reasons for excluding studies included that the study had a different trial design [41-45]; the study had a different intervention or a different control [46-64]; the study had a different population [65-67]; or the study had been registered with the Clinical Trials Registry Platform (clinicaltrials.gov or WHO ICTRP) and had been labeled “completed”, but outcomes did not report [68-73] (Supp. Table 2). 

10. The I-squared is not an evaluation of the statistical heterogeneity between summary data. It is an estimate of the percentage of variability in a meta-analysis that is attributable to study heterogeneity. 

The choice of a random vs fixed effects model should not be based on heterogeneity according to the Cochrane Handbook and other sources. For more information on choice of models see page 105 of https://www.meta-analysis.com/downloads/Intro_Models. pdf.

Answer: We appreciate your generous help for the references. In the revised manuscript, a random-effects model was used. However, if the number of studies is very small, the statistical power will have poor precision due to the variance between studies. Although the random-effects model is still the appropriate model, the information to apply it correctly is not available. In this case, we added the separate effects to our manuscript. If heterogeneity was identified (I2 >40%) and there were sufficient trials included in the review, we planned to investigate heterogeneity in the specified subgroups based on types of prokinetics (erythromycin, metoclopramide or other prokinetics), combination of prokinetics (yes or no), and feeding intolerance history (participants with or without pre-existing feeding intolerance before the start of the trial). Analysis was performed to assess whether the difference between the subgroups was statistically significant. 

Thanks again for your help and your advice. With the help of American Journal Experts (AJE), we corrected errors in grammar and spelling in our manuscript. We also hope that the revised manuscript can meet with your approval. 

Reviewers' comments:

Reviewer #1: 

Thanks a lot for your very considerate advice. In the revised manuscript, there were some substantially modified in our results and conclusions sections. we removed the results of the combining herbal group in the meta-analysis. A random-effects model was used. However, if the number of studies is very small, the statistical power will have poor precision due to the variance between studies. Although the random-effects model is still the appropriate model, the information to apply it correctly is not available. In this case, we added the separate effects to our manuscript. In addition, with the help of American Journal Experts (AJE), we corrected errors in grammar and spelling in our manuscript. 

we will reply your comments one by one in details.

1. Paragraph 115-118 needs revision. Fragment?

Answer: yes, we have amended the relevant part in manuscript.

In the revised essay, it is shown as follows: if the gastric feeding patients with feeding intolerance had a GRV ≥500 mL and/or symptoms of nausea, vomiting, abdominal distention, regurgitation, deterioration in hemodynamics or other symptoms resulting in feeding interruption, and if they failed to respond to intervention, then, regardless of whether they were in the control group or the prokinetics group, they were switched to postpyloric feeding or had gastric feeding withheld for 4-6 h.

2. Line 142: end of sentence “changes of electrocardiogram QTc interval, and so on”. Perhaps removing “and so on” will sound better or can refer to outcomes listed in the paragraph of “inclusion criteria”.

Answer: according to your advice, we have listed the outcomes of reported adverse events in the paragraph of “inclusion criteria”. In the revised essay, it is showed as follows: A reported adverse event was defined as any untoward medical occurrence or unfavorable and unintended sign, including an abnormal laboratory finding, symptom, or disease (new or exacerbated), temporally associated with the use of the study medication. The reported adverse events included abnormal laboratory test results (hematology, clinical chemistry, or urinalysis) or other safety assessments (e.g., ECGs, radiological scans, or measurements of vital signs), including those that worsened from baseline, and were deemed clinically significant in the medical and scientific judgment of the investigator; exacerbation of a chronic or intermittent preexisting condition, including an increase in the frequency and/or intensity of the condition; new conditions detected or diagnosed after the administration of study medication even if they may have been present prior to the start of the study; and/or signs, symptoms, or clinical sequelae of a suspected interaction, such as, diarrhea, nosocomial pneumonia, severe sepsis, brain herniation, cardiac arrest, or changes in the electrocardiographic QTc interval. 

3. Lines 252-254: for consistency, report results fully with point estimates and 95% CIs.

Answer: Yes, in the revised manuscript, all the results were fully with point estimates and 95% CIs.

4. Lines 256-259: what do the authors mean by “the total situation”. “And there was also no significant difference in the risk of the reported adverse events between prokinetics treatment and the control group on the total situation (RR, 1.18, 95%CI, 0.79 to 1.75, P=0.42; I2=69%).

Answer: in the revised manuscript, this part had a big change in data analysis. In the outcome of reported adverse events, it is showed as follows: “Seven studies reported events that met the definition of adverse events in 757 critically ill patients [26-28, 30-32, 35]. The meta-analysis showed that there was no significant difference in the incidence of reported adverse events between the prokinetic agent group and the control group (RR 1.13, 95% CI 0.92, 1.38; P = 0.25; I2 = 0%) (Supp. Figure 2)”.

5. Lines 264: can the authors explain why they deviated from their data analysis plan of conducting publication bias assessment when 10 or more studies were available (lines 181-183) but lowered the quality of evidence because publication bias was strongly suspected?

Answer: In the revised manuscript, the meta-analysis showed that the use of prokinetic agents in critically ill patients receiving gastric feeding may reduce ICU or hospital length of stay, there was no significant difference in the incidence of reported adverse events and all-cause mortality between prokinetic agent group and control group. Although we could not assess this category reliably due to small number of eligible studies, we did not downgrade the quality of evidence for publication bias. The reasons are chiefly as follows: in view of the fact that publication bias arises when investigators fail to report studies they have undertaken (typically those that show no effect). A prototypical situation that should elicit suspicion of publication bias occurs when published evidence is limited to a small number of trials, all of which are showing benefits of the studied intervention. However, in our study, not all of included studies were showing benefits of the studied intervention in clinical outcome of ICU length of stay and hospital length of stay, respectively. 

6. Line 271-272: “Subgroup analysis by type of prokinetics.” Fragment.

Answer: yes, we have amended the relevant part in manuscript. 

it is showed as follows: “Although no significant heterogeneity was found, we performed subgroup analyses to determine if there are important subgroup differences. In the subgroup analysis stratified by type of prokinetic agents, there were no significant subgroup difference in the clinical outcomes of hospital length of stay, ICU length of stay, reported adverse events and all-cause mortality (Supp. Figure 4-7)”.

7. Line 275-277: When Metoclopramide did not show significant difference in the hospital length of stay compared with the control group (MD, 1.70, 95%CI, -4.75 to 8.15, P=0.61; I2=0%) (S1 Fig). Check grammar please.

Answer: In the revised manuscript, we did not report the outcome of subgroup analysis by specific drug. Because the number of studies is very small, the statistical power will have poor precision due to the variance between studies. We added a table of the separate effects of different prokinetics on the ICU length of stay and hospital length of stay in Supp. Table 3.

8. Lines 283-284: repetition of 254-256.

Answer: we have amended the relevant part in manuscript.

9. Lines 285-286: review grammar. 

Line 286-287: pooled effects for what outcome(s)?

Answer: we have amended the relevant part in manuscript.

10. Line 288: I think the authors mean to have the first part as a subheading “Subgroup analysis by preventive giving prokinetics or therapeutic usage of prokinetics.” Would recommend: preventive use instead of preventive giving. Also would recommend: “pooled effects for primary outcomes” instead of “pooled effects” at the end of sentences.

Answer: yes, we accept your opinions, and we have amended the relevant part in manuscript. In the revised manuscript, this part is amended as follows: Furthermore, there was no study comparing the combination of prokinetics to placebo or no treatment. Only one study about the preventive usage of prokinetics for risk patients demonstrated the outcome of reported adverse events [27]. The others were about the preventive usage of prokinetics for all patients. The subgroup analysis result of the preventive usage of prokinetics for all patients did not show important changes in the pooled effects of the reported adverse events.

11. Lines 298-301: the authors deviate from statistical plan for publication bias again by downgrading for string suspicion of publication bias. Revision needed. Also in GRADE evidence profile confidence intervals for all-cause mortality, ICU and hospital length of stay and adverse events all include significant benefits and harms and yet no downgrading for imprecision was performed. Can the authors elaborate on this ? ¬¬¬For length of stay what was the unit of measurement? Days or hours? This would be important for the outcome of hospital length of stay with natural medicines. If the difference was 9 days fewer and between 18.5 and 0.8 days lower, one might consider downgrading for imprecision specially that the number of patients in that analysis is 122 only. I would strongly recommend the authors consult someone with more experience using the GRADE method.

Answer: we felt sorry for these mistakes. We seriously conducted the GRADE evaluation again. The result of GRADE is amended as follows: The certainty of evidence was moderate for the clinical outcome of all-cause mortality. However, the certainty of evidence was low for the clinical outcomes of ICU length of stay, hospital length of stay and reported adverse events. The details of the risk of bias and quality assessment are outlined in Table 3. The rationale for decisions had been made clear through footnotes to Table 3. 

12. Lines 315-318: And there was a systematic review about the definition, prevalence, and outcome of feeding intolerance in intensive care, and the result showed that there were more than 60% (40/63) of studies defining the large GRVs with the threshold less than or equal to 250 ml and about 41.7% (30/72) of studies with the sole GRVs levels to definite of feeding intolerance. Not clear and needs to be rewritten. 

Answer: yes, we have amended the relevant part in manuscript. The revised sentences are as follows: There are three methods for the treatment of gastric feeding intolerance. First, there is the most widely used method, the administration of prokinetics. Among recipients of gastric feeding, 13% had been prescribed prokinetics preemptively before they developed intolerance. Approximately one-third of patients who developed feeding intolerance were treated with a prokinetic agent during their stay in the ICU. Second, after the development of intolerance, 17% of patients received supplemental parenteral nutrition. Third, only 7.5% of patients with gastric feeding intolerance subsequently received enteral nutrition via a postpyloric feeding tube [8]. (In the revised essay, the revised sentences are in the part of “Introduction”).

13. Lines 325-327: So, we conducted this updated systematic review and meta-analysis, we defined the gastric enteral nutrition feeding intolerance as either GRV ≥500 ml or GRV <500 ml concomitant with symptoms of nausea, vomiting, abdominal distention, regurgitation or other symptoms resulting in feeding interruption. This is not true. Only 3 of the included studies explicitly reported the cutoff for the GRV to be ≥ 500 ml (24, 31, 34) and 1 study reported > 200 ml at any time or > 500 ml/24 hours). I find this misleading. 

Line 342: Lewis, K. et al [12] defined feeding intolerance as either GRV ≥150 ml, vomiting, or abdominal distention resulting in feeding interruption. This definition may be considered obsolete [13]. This is very difficult to justify given the point above when the exact amount of GRV was not reported in more than half of the studies included.

Answer: yes, we fully understand your purpose. We need to clarify the aims and objectives of our study. In this study, except for feeding tolerance, we also examined the effect of prokinetic agents on gastrointestinal symptoms and clinical outcomes. We defined gastric feeding intolerance as either GRV ≥500 ml or concomitant with symptoms of nausea, vomiting, abdominal distention, regurgitation or other symptoms resulting in feeding interruption in critically ill adult patients receiving gastric feeding tubes. We excluded studies that discontinued or interrupted gastric feeding prematurely following the disappearance of gastric feeding intolerance. Under this latest definition, our meta-analysis found some new studies [26-30, 33, 34], besides, we identified 5 studies regarding the administration of prokinetics of herbal medicines/natural medicines in patients receiving gastric feeding tube in critically ill adults [28-30, 37, 40]. 

14. Lines 342-347 are exact repetition of lines 324-329. Line 351: “trails” change to “trials”

Answer: Thank you for your warm reminder. We improve the quality of the English throughout our manuscript with the help of American Journal Experts (AJE). 

15. Lines 352-358: I disagree with the conclusion. The heterogeneity was 95%, the confidence intervals we extremely wide (especially if this was in days) and the total number of patients is 122. This makes it extremely difficult to make any sound conclusions, contrary to the way reported by the authors.

Answer: In the revised manuscript, we were careful to generalize the conclusion of this article. we removed the results of the combining herbal group in the meta-analysis. “we found that prokinetic agents might reduce ICU or hospital length of stay in critically ill patients receiving gastric feeding. However, the number of studies and the sample size were very small, and the certainty of evidence was low. Furthermore, there was no significant difference between prokinetic agent groups and placebo/no treatment in the risk of reported adverse events and all-cause mortality. Therefore, we cannot draw a convincing conclusion that the use of prokinetics could improve clinical outcomes in critically ill adults. We recommend a more comprehensive search and further original studies on this topic”.

16. Lines 380: I am concerned that the while the title of the, manuscript is prokinetics, the bulk of discussion section and the conclusion is about herbal medicines as prokinetic agents. As mentioned earlier, the authors claim their cutoff for GRV was 500 ml but they included many studies without a clear cutoff and by doing that they missed many studies in Lewis et al. that included generic medicines. In addition, the authors did not include hospital length of stay as an outcome in their methods section (lines 92-101) and yet they perform this analysis and include the forest plot for it in the main manuscript.

The methods section very well be written. However, the other sections need significant revision of language and grammar.

Answer: your comment on the methods section is a very important recognition for our manuscript. The revised portion are marked with the changes highlighted in yellow in the paper. We hope the revised manuscript meet you’re your approval.

Reviewer #2: 

Thank you for your warm reminder. Your suggestions and ideas have been highly regarded. We will reply your comments with further instructions.

The authors made a new meta-analysis on the efficacy and safety of prokinetic agents administered during enteral feeding in intensive care patients. Previous meta-analysis was already performed on this topic. Originality of this work may come from a revised definition for digestive feeding intolerance, which indeed corresponds to more recent experts’ recommendations (Gastric residual volume (GRV) above 500mls). It has to be noted however, that even the threshold of 500mls GRV for definition of gastric enteral feeding intolerance is challenged, and some authors now recommend against measuring any GRV during enteral nutrition in ICU patients.

I have two major concerns with that study:

1. Subgroup analysis: what is the rationale of pooling studies in which different substances are administered: indeed, natural/herbal substances are not identical in these pooled studies.

Answer: first, some studies have suggested that measurement of GRV provides no benefit and should no longer be recommended. However, GRV is also an indicator of feeding intolerance in many ICUs, especially in patients with a high risk of aspiration and aspiration pneumonia. Therefore, the Chinese guidelines call for caution in abandoning monitoring of GRV in some high-risk patients. 

Second, after careful consideration, we agree with your advice, the natural/herbal substances are not identical in these pooled studies. In the revised manuscript, we have removed the pooled effect of the natural/herbal substances. Besides, although no significant heterogeneity was found, we performed subgroup analyses to determine if there are important subgroup differences. In the subgroup analysis stratified by type of prokinetic agents, there were no significant subgroup difference in the clinical outcomes of hospital length of stay, ICU length of stay, reported adverse events and all-cause mortality (Supp. Figure 4-7). 

We found that prokinetic agents might reduce ICU or hospital length of stay in critically ill patients receiving gastric feeding. However, the number of studies and the sample size were very small, and the certainty of evidence was low. Furthermore, there was no significant difference between prokinetic agent groups and placebo/no treatment in the risk of reported adverse events and all-cause mortality. Therefore, we cannot draw a convincing conclusion that the use of prokinetics could improve clinical outcomes in critically ill adults. We recommend a more comprehensive search and further original studies on this topic.

2. To my opinion, the conclusion (administration of herbal medicines/natural medicines may reduce hospital length of stay) is not supported by the (meta)analysis. The purpose of a meta-analysis is to pool pertinent studies in order to answer a well predefined question. The conclusion that is drawn by the authors is not really related to the initial question.

Answer: we amended the relevant part in manuscript carefully. Revised portion are marked with the changes highlighted in yellow in the paper. there were some substantially modified in our results and conclusions sections. 

In the revised manuscript, the conclusion was showed as follows: As a class of drugs, prokinetic agents may improve gastric feeding tolerance in critically ill adults. However, there is low certainty in the evidence that prokinetic agents are effective in reducing ICU or hospital length of stay. There was also no significant reduction in the risk of reported adverse events and all-cause mortality. Additional RCTs are needed to determine the effect of prokinetics on clinical outcomes in critically ill patients in the future.

Thank you very much for your attention and kindly advise. We hope the revised manuscript meet with your approval.

---

## [Decision Letter · Decision Letter 1]

9 Jun 2020

PONE-D-20-01167R1

The efficacy and safety of prokinetics in critically ill adults receiving gastric feeding tubes: a systematic review and meta-analysis

PLOS ONE

Dear Dr. Zhang,

Thank you for submitting your manuscript to PLOS ONE. After careful consideration, we feel that it has merit but does not fully meet PLOS ONE’s publication criteria as it currently stands. Therefore, we invite you to submit a revised version of the manuscript that addresses the points raised during the review process.

The review is greatly improved in clarity and transparency, but requires some minor changes. Please address the comments of the reviewers and in addition:

For Line 288-289 where you state that hospital LOS was not significant I suggest you revise to use the same language as in 292-3 for ICU LOS, in that there appears to be a positive effect, unless you wish to clarify in what way the hospital LOS was not significant (clinically?)For the outcomes of gastrointestinal symptoms and feeding tolerance, you should mention that if outcomes could not be combined by meta-analysis you summarized them narratively. You only discuss in the methods how you will use meta-analysis and then you do not meta-analyse the symptoms and tolerance outcomes, I assume because they are not appropriate to meta-analyze.For Line 301 and the outcome of adverse events it is preferable to use the term 'risk' rather than 'incidence' as incidence implies measurement of time at risk.For Table 3 please include the units for each outcome, e.g., days and deathsFor all forest plots, including Fig 2, Fig 3, S2Fig and S3Fig, please specify the comparison and the outcome (with units) in the header and replace the bracketed experimental and control on the x axis with a legend indicating the comparisons.

We look forward to receiving your revised manuscript.

Kind regards,

Lisa Susan Wieland

Academic Editor

PLOS ONE

Reviewers' comments:

Reviewer's Responses to Questions

**Comments to the Author**

1. If the authors have adequately addressed your comments raised in a previous round of review and you feel that this manuscript is now acceptable for publication, you may indicate that here to bypass the “Comments to the Author” section, enter your conflict of interest statement in the “Confidential to Editor” section, and submit your "Accept" recommendation.

Reviewer #1: (No Response)

Reviewer #3: (No Response)

2. Is the manuscript technically sound, and do the data support the conclusions?

Reviewer #1: Yes

Reviewer #3: Yes

3. Has the statistical analysis been performed appropriately and rigorously? 

Reviewer #1: Yes

Reviewer #3: I Don't Know

4. Have the authors made all data underlying the findings in their manuscript fully available?

Reviewer #1: Yes

Reviewer #3: Yes

5. Is the manuscript presented in an intelligible fashion and written in standard English?

Reviewer #1: Yes

Reviewer #3: Yes

6. Review Comments to the Author

Reviewer #1: I thank the authors for the significant work done to address al comments and i find the reviewed submission substantially improved.

the length of stay unit for hospital and ICU is still not clear. I assume it is days. However would be nice to have it clarified. See my original comment #11.

line 377: "We recommend a more comprehensive search and further original studies on this topic." i recommend the words "more comprehensive search" be deleted as they give the impression the authors did not perform a comprehensive search.

Reviewer #3: Thank you for the opportunity to review the revised manuscript. In this systematic review, authors evaluated the effect of prokinetics in critically ill adults on gastric feeding tube tolerance according to the updated definition. This systematic review implies that prokinetics improves tolerance of enteral feeding, and additionally provides the attractive hypothesis that prokinetics may shorten the length of ICU and hospital stay. Although authors tried to perform meta-analysis about gastric feeding tube tolerance, study diversity (e.g. various interventions and various outcome definitions) did not allow the authors data synthesis. Authors seems to revise their manuscript well according to the previous editor's and reviewers' comments.

Comments to the authors:

1. As authors state in background, the aim of this study is to evaluate the effect of prokinetics on gastric feeding tube tolerance. So, the main results of this study is the description about this effect (L273-283), not about ICU and hospital length of stay. And one of key points of this study, I believe, is the difficulty to compare results across previous studies because of various outcome definitions, and necessity of the valid measure of gastric tube tolerance in future studies. Authors should add more concise description in this paragraph (L273-283) to show the potential benefit on gastric feeding tube tolerance and clarify the abovementioned point.

2. L288-290: "Those five studies, enrolling a total of 250 patients, demonstrated that there was no significant difference in hospital length of stay ..."

Are there any significant difference between groups about the hospital length of stay? 95%CI of -5.35 to -1.06 is significant, isn't it? Please check.

7. PLOS authors have the option to publish the peer review history of their article (what does this mean?). If published, this will include your full peer review and any attached files.

Reviewer #1: No

Reviewer #3: Yes: Kyohei Miyamoto

---

## [Author Response · Author response to Decision Letter 1]

26 Jun 2020

Dear Editors and Reviewers, 

Thank you for your letter and the comments concerning our manuscript (Manuscript Number: PONE-D-20-01167R1). We appreciate your positive comments regarding our manuscript. Your suggestions and ideas have been carefully considered. Revised portions are marked with changes in colored fonts in the paper. We hope that the revised manuscript will meet with your approval. The main corrections in the paper and our responses to the reviewers’ comments are as follows:

Editors' comments:

The review is greatly improved in clarity and transparency, but requires some minor changes. Please address the comments of the reviewers and in addition:

1. For Line 288-289 where you state that hospital LOS was not significant I suggest you revise to use the same language as in 292-3 for ICU LOS, in that there appears to be a positive effect, unless you wish to clarify in what way the hospital LOS was not significant (clinically?)

Answer: We sincerely appreciate your thoughtful advice. This mistake was due to our carelessness in writing, and all authors sincerely apologize for this mistake. We have corrected this error. In the revised paper, the text is as follows: “These five studies, which enrolled a total of 250 patients, demonstrated a significant difference in the hospital length of stay between the prokinetic agent-treated group and the control group (MD -3.21, 95% CI -5.35, -1.06; P = 0.003; I2 = 28%) (Fig 2).”

2. For the outcomes of gastrointestinal symptoms and feeding tolerance, you should mention that if outcomes could not be combined by meta-analysis you summarized them narratively. You only discuss in the methods how you will use meta-analysis and then you do not meta-analyse the symptoms and tolerance outcomes, I assume because they are not appropriate to meta-analyze.

Answer: Yes. The various outcome definitions, especially for gastric tube tolerance, precluded quantitative synthesis of the data. According to your advice, we have amended this part in the revised manuscript as follows: “Thirteen studies evaluated the effect of prokinetics on gastrointestinal symptoms and/or feeding tolerance in adult critically ill patients receiving gastric feeding [26-30, 32-34, 36-40]. The main results obtained are as follows: gastric emptying, GRV, diarrhea, constipation, feeding complications and feeding intolerance. Gastric emptying was measured by the drug model of acetaminophen absorption or the 13C-octanoic acid breath test with calculation of the gastric emptying time, gastric emptying coefficient or area under the plasma concentration-time curve. The various outcome definitions, especially for gastric tube tolerance, precluded quantitative synthesis of the data.”

3. For Line 301 and the outcome of adverse events it is preferable to use the term 'risk' rather than 'incidence' as incidence implies measurement of time at risk.

Answer: Yes. Following your suggestion, to be more accurate, we have replaced “incidence” with “risk”.

4. For Table 3 please include the units for each outcome, e.g., days and deaths

Answer: Thank you for this comment. We have revised the manuscript to include the unit for each outcome in Table 3. We hope that this change improves the readability of the data. 

5. For all forest plots, including Fig 2, Fig 3, S2Fig and S3Fig, please specify the comparison and the outcome (with units) in the header and replace the bracketed experimental and control on the x axis with a legend indicating the comparisons.

Answer: Following your suggestion, we have added “units” and “legend” to each forest plot. These changes have been made to the text to improve the readability and to clarify the interpretation of the data.

Reviewers' comments:

Reviewer #1: 

1. I thank the authors for the significant work done to address al comments and i find the reviewed submission substantially improved.

Answer: Thank you for your very considerate advice; your positive comment on our manuscript is sincerely appreciated. We will reply to your comments one by one in detail.

2. the length of stay unit for hospital and ICU is still not clear. I assume it is days. However would be nice to have it clarified. See my original comment #11.

Answer: We apologize for this mistake. The unit for the lengths of hospital stay and ICU stay is “days”. We have added “units” in Table 3 in the revised manuscript. We hope that these changes improve the readability and clarify the interpretation of the data.

3. line 377: "We recommend a more comprehensive search and further original studies on this topic." i recommend the words "more comprehensive search" be deleted as they give the impression the authors did not perform a comprehensive search.

Answer: Thank you for your thoughtful reminder. We agree with your advice; the words "more comprehensive search" have been deleted in the revised manuscript.

Reviewer #3: 

Thank you for the opportunity to review the revised manuscript. In this systematic review, authors evaluated the effect of prokinetics in critically ill adults on gastric feeding tube tolerance according to the updated definition. This systematic review implies that prokinetics improves tolerance of enteral feeding, and additionally provides the attractive hypothesis that prokinetics may shorten the length of ICU and hospital stay. Although authors tried to perform meta-analysis about gastric feeding tube tolerance, study diversity (e.g. various interventions and various outcome definitions) did not allow the authors data synthesis. Authors seems to revise their manuscript well according to the previous editor's and reviewers' comments.

Comments to the authors:

1. As authors state in background, the aim of this study is to evaluate the effect of prokinetics on gastric feeding tube tolerance. So, the main results of this study is the description about this effect (L273-283), not about ICU and hospital length of stay. And one of key points of this study, I believe, is the difficulty to compare results across previous studies because of various outcome definitions, and necessity of the valid measure of gastric tube tolerance in future studies. Authors should add more concise description in this paragraph (L273-283) to show the potential benefit on gastric feeding tube tolerance and clarify the abovementioned point.

Answer: Thank you for your thoughtful reminder. These comments are valuable and very helpful for revising and improving our paper and provided important guiding significance for our research. According to your advice, we have amended this part in the revised manuscript as follows: 

“Thirteen studies evaluated the effect of prokinetics on gastrointestinal symptoms and/or feeding tolerance in adult critically ill patients receiving gastric feeding [26-30, 32-34, 36-40]. The main results obtained are as follows: gastric emptying, GRV, diarrhea, constipation, feeding complications and feeding intolerance. Gastric emptying was measured by the drug model of acetaminophen absorption or the 13C-octanoic acid breath test with calculation of the gastric emptying time, gastric emptying coefficient or area under the plasma concentration-time curve. The various outcome definitions, especially for gastric tube tolerance, precluded quantitative synthesis of the data.

As a class of drugs, prokinetic agents appear to have positive effects on gastrointestinal function and improving feeding tolerance. Ten of the thirteen studies reported positive effects on improving gastric emptying and/or resolution of feeding intolerance in critically ill patients with the use of prokinetic agents. However, two studies suggested that metoclopramide had no effect on decreasing gastrointestinal complications in adult neurocritical patients or critical traumatic brain injury patients. One study reported that rikkunshito did not improve the achievement of enteral calorie targets in critically ill adults (Table 2).”

2. L288-290: "Those five studies, enrolling a total of 250 patients, demonstrated that there was no significant difference in hospital length of stay ..."

Are there any significant difference between groups about the hospital length of stay? 95%CI of -5.35 to -1.06 is significant, isn't it? Please check. 

Answer: I apologize for this mistake. We have corrected this error. In the revised manuscript, the text is as follows: “These five studies, which enrolled a total of 250 patients, demonstrated a significant difference in the hospital length of stay between the prokinetic agent-treated group and the control group (MD -3.21, 95% CI -5.35, -1.06; P = 0.003; I2 = 28%) (Fig 2)”.

Thank you again for your attention and thoughtful advice. We hope that the revised manuscript will meet with your approval.

---

## [Decision Letter · Decision Letter 2]

17 Nov 2020

PONE-D-20-01167R2

The efficacy and safety of prokinetics in critically ill adults receiving gastric feeding tubes: A systematic review and meta-analysis

PLOS ONE

Dear Dr. Zhang,

Thank you for submitting your manuscript to PLOS ONE. After careful consideration, we feel that it has merit but does not fully meet PLOS ONE’s publication criteria as it currently stands. Therefore, we invite you to submit a revised version of the manuscript that addresses the points raised during the review process.

Please address the comments from the reviewer on the attachment, and consider shortening the introduction as requested.

We look forward to receiving your revised manuscript.

Kind regards,

Lisa Susan Wieland

Academic Editor

PLOS ONE

Reviewers' comments:

Reviewer's Responses to Questions

**Comments to the Author**

1. If the authors have adequately addressed your comments raised in a previous round of review and you feel that this manuscript is now acceptable for publication, you may indicate that here to bypass the “Comments to the Author” section, enter your conflict of interest statement in the “Confidential to Editor” section, and submit your "Accept" recommendation.

Reviewer #3: All comments have been addressed

Reviewer #4: All comments have been addressed

2. Is the manuscript technically sound, and do the data support the conclusions?

Reviewer #3: (No Response)

Reviewer #4: Yes

3. Has the statistical analysis been performed appropriately and rigorously? 

Reviewer #3: (No Response)

Reviewer #4: Yes

4. Have the authors made all data underlying the findings in their manuscript fully available?

Reviewer #3: (No Response)

Reviewer #4: Yes

5. Is the manuscript presented in an intelligible fashion and written in standard English?

Reviewer #3: (No Response)

Reviewer #4: Yes

6. Review Comments to the Author

Reviewer #3: (No Response)

Reviewer #4: Minor clarifications line (see attached):

36

73

355

Also the introduction should be made more concise in this long paper.

7. PLOS authors have the option to publish the peer review history of their article (what does this mean?). If published, this will include your full peer review and any attached files.

Reviewer #3: **Yes: **Kyohei Miyamoto

Reviewer #4: No

---

## [Author Response · Author response to Decision Letter 2]

6 Dec 2020

Dear Editors and Reviewers,

Thank you for your letter and the comments concerning our manuscript (Manuscript Number: PONE-D-20-01167R2). We appreciate your positive comments on our manuscript. Your suggestions and ideas have been carefully considered. The revised portions are in red font in the revised version of the paper. We hope the revised manuscript will meet with your approval. The main corrections in the paper and the responses to the reviewers’ comments are as follows:

Editors' comments:

Please address the comments from the reviewer on the attachment, and consider shortening the introduction as requested.

Answer: We are very appreciative of your advice. We have replied to the reviewers' comments in a point-by-point manner.

Reviewers' comments:

Reviewer #3:

1. If the authors have adequately addressed your comments raised in a previous round of review and you feel that this manuscript is now acceptable for publication, you may indicate that here to bypass the “Comments to the Author” section, enter your conflict of interest statement in the “Confidential to Editor” section, and submit your "Accept" recommendation.

Reviewer #3: All comments have been addressed

Answer: We appreciate your positive comments on our manuscript. Thank you for your hard work reviewing our manuscript.

Reviewer #4:

1. If the authors have adequately addressed your comments raised in a previous round of review and you feel that this manuscript is now acceptable for publication, you may indicate that here to bypass the “Comments to the Author” section, enter your conflict of interest statement in the “Confidential to Editor” section, and submit your "Accept" recommendation.

Reviewer #4: All comments have been addressed

2. Is the manuscript technically sound, and do the data support the conclusions?

Reviewer #4: Yes

3. Has the statistical analysis been performed appropriately and rigorously?

Reviewer #4: Yes

4. Have the authors made all data underlying the findings in their manuscript fully available?

Reviewer #4: Yes

5. Is the manuscript presented in an intelligible fashion and written in standard English?

Reviewer #4: Yes

6. Review Comments to the Author

Reviewer #4: Minor clarifications line (see attached):

36

73

355

Also the introduction should be made more concise in this long paper.

Answer: We appreciate your positive comments on our manuscript. Your comments were valuable when we revised and improved our paper. According to your advice, we have amended those parts in the revised manuscript. We will reply to your comments one by one in detail.

（1）Line 36: Equivocal: Continuous gastric feeding is not physiological, though possibly safer than bolus feeding (more physiological) regarding potential aspiration. Continuous intestinal feeding may be therefore actually more physiological.

Answer: We agree with you. We apologize that we did not express ourselves very clearly. We would like to present our standpoint that feeding administered in the stomach is usually considered more physiological than postpyloric feeding.

（2）Line 73: These sentences: “Some studies have suggested that measurement of GRV provides no benefit and should no longer be recommended. However, GRV is also an indicator of feeding intolerance in many ICUs.” were deleted.

Answer: Following your suggestion, we have deleted these sentences.

（3）Line 355: The negative studies (3 of 13, 23.08%) were hampered by special populations of neuro-critical patients and critical traumatic brain injury patients taking metoclopramide and by the use of the specific drug rikkunshito.

Answer: I am very sorry that we did not state it clearly. We have amended this part in the revised manuscript. It is shown as follows: “The studies that did not show beneficial effects (3 of 13, 23.08%) investigated special populations of neurocritical patients and critical traumatic brain injury patients taking metoclopramide and rikkunshito.”

（4）Also the introduction should be made more concise in this long paper.

Answer: Thank you for your reminder. These comments were very helpful for revising and improving our paper. According to your advice, we have amended this part in the revised manuscript.

Thank you again for your attention and advice. We hope the revised manuscript will meet with your approval.

---

## [Editor Report · Decision Letter 3]

29 Dec 2020

The efficacy and safety of prokinetics in critically ill adults receiving gastric feeding tubes: A systematic review and meta-analysis

PONE-D-20-01167R3

Dear Dr. Zhang,

We’re pleased to inform you that your manuscript has been judged scientifically suitable for publication and will be formally accepted for publication once it meets all outstanding technical requirements.

Kind regards,

Lisa Susan Wieland

Academic Editor

PLOS ONE
---

## [Editor Report · Acceptance letter]

2 Jan 2021

PONE-D-20-01167R3 

The efficacy and safety of prokinetics in critically ill adults receiving gastric feeding tubes: A systematic review and meta-analysis 

Dear Dr. Zhang:

I'm pleased to inform you that your manuscript has been deemed suitable for publication in PLOS ONE. Congratulations! Your manuscript is now with our production department. 

Kind regards, 

on behalf of

Dr. Lisa Susan Wieland 

Academic Editor

PLOS ONE